# Job Demands and Exhaustion in Firefighters: The Moderating Role of Work Meaning. A Cross-Sectional Study

**DOI:** 10.3390/ijerph18189819

**Published:** 2021-09-17

**Authors:** Andra Cătălina Roșca, Alexandru Mateizer, Cristina-Ioana Dan, Evangelia Demerouti

**Affiliations:** 1Department of Sociology, Faculty of Political Sciences, National University of Political Studies and Public Administration, 012104 Bucharest, Romania; 2Department of Psychology, Faculty of Psychology and Educational Sciences, University of Bucharest, 050663 Bucharest, Romania; alexandru.mateizer@gmail.com; 3Centre for Psychosociology, Ministry of Internal Affairs, 023975 Bucharest, Romania; cristinaioanadan@yahoo.com; 4Department of Industrial Engineering & Innovation Sciences, Eindhoven University of Technology, Atlas Building, Room 7.405, 5600 MB Eindhoven, The Netherlands; e.demerouti@tue.nl

**Keywords:** JD-R, work meaning, exhaustion, job demands, firefighters, buffer

## Abstract

Emotional exhaustion and other symptoms of burnout are often found among emergency services professions, such as firefighting. Given the social importance of this activity and the high responsibility it requires, prevention and alleviation of burnout symptoms become primary concerns in ensuring the well-being of firefighters. Although work meaning is one of the factors associated with a lower risk of developing burnout, its protective role has not been studied in firefighters. Therefore, the purpose of this study was to assess the buffering role of work meaning in the health-impairment process of the Job Demands-Resources model, targeting the relationship between job demands and related emotional exhaustion. A cross-sectional survey design was used to collect data from a sample consisting of Romanian firefighters (*n* = 1096). Structural equation modeling indicated a positive link between job demands and exhaustion. In addition, deriving personal meaning from work was associated with lower levels of exhaustion in firefighters. A small but significant interaction effect between work meaning and job demands showed that higher levels of work meaning attenuated the positive relationship between job demands and exhaustion. In conclusion, our findings suggest that work meaning has a buffering effect on the impact of various job demands on job-related exhaustion. Nevertheless, the small effect sizes warrant further research on this topic.

## 1. Introduction

Interest in research regarding factors that may contribute, prevent, or alleviate occupational burnout has been long-standing given the multi-level implications of this phenomenon and its pervasiveness across various work-related contexts. Despite the high level of strain [1], emergency services professions, such as firefighting, are fundamental in the maintenance of complex societies and have an intrinsic positive meaning and rewarding aspects for both the professionals involved and the beneficiaries. A study [2] on a sample of 101 French firefighters found that job characteristics related to heavy physical work, time pressures, and emotionally demanding situations were associated with emotional exhaustion and other manifestations of burnout syndrome. The study concluded that the work of firefighters appears to be a strong source of stress and mental strain. Because of the high responsibility and probability of exposure to traumatic experiences characteristic of firefighters’ work, knowledge about the prevention of psychological phenomena that contribute to burnout syndrome is crucial in ensuring the well-being of firefighters and the outcomes of their service.

Perceiving one’s work as meaningful has been found to be associated with increased well-being [3,4,5,6,7] by reducing the impact of work-related stress [8] and adding to the sense of purpose in life [9]. In addition, studies have shown work meaning to be an important driver of motivation [5,10,11] associated with a lower risk of developing burnout [8,12,13,14]. Although previously studied, mostly in health-care settings, the potential protective role of work meaning in relation to the effects of prolonged exposure to stressful situations has not been investigated in firefighters. By employing the Job Demands-Resources (JD-R) model [15] and the Self-determination theory (SDT) [16], the objective of this study was to assess the impact of work meaning in the context of firefighters’ job demands and job-related exhaustion.

The strain process in firefighters was often studied within the JD-R framework [15], in which it was postulated that, in certain conditions, job demands classified as challenging (e.g., task complexity, workload, and responsibility) and hindering (e.g., bureaucracy, role conflict, and hassles) [17,18], may initiate a health-impairment process that can lead to chronic exhaustion and physical health issues [19]. In return, negative strain can determine a higher intensity of perceived job demands, fueling a loss-cycle of high job demands and negative strain, leading to burnout. High-risk professions, such as firefighting, introduce specific demands that include exposure to emotional and physical trauma and life-threatening situations on a regular basis and have a high impairment potential. Despite the fact that resilience factors in firefighters are higher than in the general population [20], previous studies have shown that the particular nature of firefighters’ job demands increases the risk for post-traumatic stress [21,22], burnout [23,24,25], and mental and physical health problems [26,27,28,29]. As such, employing the JD-R model, we expected that our data would support these results, and we hypothesized a positive relationship between the perceived intensity of job demands and exhaustion levels (Hypothesis 1).

However, the engagement of organizational and personal resources has been found to buffer the impact of job demands on negative strain [19,30,31,32], as proposed in the JD-R model. According to the authors of [19], personal resources, such as optimism and self-efficacy, facilitate more effective means of handling challenging and hindering job demands. Their studies also indicate that job resources can have a positive impact on motivation and engagement, especially in situations in which job demands are high [33,34]. It is within this framework that we introduce the concept of work meaning and the possibility of its protective role against burnout symptoms in firefighters. For the purpose of this article, we use Steger’s [5] definition of work meaning as the subjective experience of one’s own work as having positive value, positive social impact, and enabling personal growth. Experiencing meaning at work is not only associated with the search for meaning in a static person-job fit sense, but rather with a striving and a dynamic process that implies self-actualization, adaptation, and construction of meaning. From this perspective, the ability to construct meaning becomes a personal resource, especially in dynamic environments such as changing organizations [35]. Studies taking different approaches to the concept of work meaning have found that it is linked to a series of positive outcomes for both the individual and the organization, such as job performance [4,36,37], organizational commitment [38,39,40], employee engagement [37,41,42,43], motivation [5,10,11], and well-being [3,4,5,6,7]. By contrast, a poor sense of meaning in the workplace has been associated with negative outcomes, particularly employee burnout symptoms [12,38]. Relevant to our sample, in his exploratory study on firefighters, the author of [44] found that different dimensions of meaning in life, a close correlate of work meaning [3,9], had beneficial effects against several dimensions of burnout, such as less emotional exhaustion and a greater sense of personal accomplishment.

Finding meaning in work implies a satisfactory match between the sense of existential purpose and the activities one performs. As such, work becomes a domain that facilitates the acting out of an individual’s intended role, accompanied by the experience of autonomy, competence, and relatedness. As proposed by the self-determination theory (SDT) [16], this situation fosters the most high-quality forms of motivation and engagement in activities, sustaining voluntary efforts, persistence, and enhanced performance. The intrinsic motivation characteristic of meaningful work decreases the sense of work as an externally imposed burden and the sense of discontinuity between the self and the work role. In addition, SDT states that distortions in this setting have a robust detrimental effect on a person’s well-being. On this note, some authors [12] suggest that certain organizational policies, such as performance-contingent incentives, applied in fundamental prosocial professions (e.g., medicine), may deplete work of its meaning and, in doing so, may undermine practitioners’ intrinsic motivation, autonomy, competency, and the sense of calling and increase the odds of developing burnout symptoms. Given these considerations, we expect to find a negative relationship between work meaning and emotional exhaustion within our firefighter sample (Hypothesis 2).

Additionally, considering the job characteristics of firefighters, namely that engagement and dedication are crucial aspects in working for a greater good, we expect work meaning to play a significant role in their attitude towards the various job demands they encounter. According to some authors [17], although job demands, both challenging and hindering, are related to exhaustion and burnout, challenging demands, such as high responsibility and time pressures, also have the capacity to trigger positive emotions and promote engagement, personal growth, and mastery. On the same note, studies conducted in the JD-R framework have found that personal resources can help employees better deal with hindering job demands, such as bureaucracy and role conflicts [19]. Deriving a high sense of meaning from work and thus maintaining drive could enable individuals to rely more on their personal and job resources in order to manage the various stressful situations related to their work. By focusing on their important social role, on work as a source of personal meaning and growth, and on the significance of their group identity, firefighters might enrich their personal resources and find effective ways to deal with their job requirements. From this perspective, we hypothesize that work meaning will buffer the impact of challenging and hindering job demands on negative strain (Hypothesis 3).

Although the positive role of work meaning and its inverse association with burnout symptoms have been documented in different professions, such as physicians and nurses [14], pediatric residents [13], oncologists [45], schoolteachers [46], and social workers [47], there are still aspects of this relationship that need further investigation. How work meaning is maintained or constructed under various organizational contexts is an important issue that can add substantially to the understanding of this concept and its outcomes and interactions with other factors. For example, some researchers [48] found that, in some cases, meaningful work can also be a risk factor for exhaustion as a consequence of overly increased efforts in situations in which perceived obligations related to an organization’s mission, values, and principles are transgressed (ideological psychological contract breach) [49]. However, despite the complexities that may arise, we expect that under usual circumstances, work meaning retains its positive impact.

In brief, the present study is focused on the protective role of work meaning in relation to emotional exhaustion in a sample of Romanian firefighters. We expect job demands to be a risk factor for exhaustion and to find a buffering effect introduced by work meaning in this dynamic.

## 2. Materials and Methods

### 2.1. Participants and Procedure

We conducted our research based on a sample of 1096 firefighters from 27 Romanian fire departments. Recruitment and the questionnaire application process, along with the informed consent of the participants, were carried out by unit psychologists without the researchers’ direct participation. They emphasized the confidentiality and anonymity aspects of participation and motivated respondents to provide sincere and open responses. The selection process included all types of interventions assigned to Romanian firefighter units. In our sample, 99.4% of respondents were men, and 0.6% were women. The mean age of the participants was 38.47 years (SD = 7.04; range: 20–58 years). A total of 72.5% of the participants had a non-leadership position, and 27.5% had a leadership position. In addition, 61.7% held a high school diploma and 38.3% had a university degree. The majority of participants were married or in a relationship (77.6%), while 22.4% were single or divorced. Regarding the level of seniority, the majority of participants (73.7%) had been in service for 10 to 20 years.

The study was conducted according to the guidelines of the Declaration of Helsinki and approved by the Institutional Review Board (or Ethics Committee) of NUPSPA (protocol code no. 120, 1 June 2020).

### 2.2. Measures

Work meaning was measured with the Work and Meaning Inventory (WAMI) [5], a self-report instrument that includes three dimensions: positive meaning (e.g., “I have found a meaningful career.”), consisting of four items; meaning-making (e.g., “I view my work as contributing to my personal growth.”); and greater good motivations (e.g., “I know my work makes a positive difference in the world.”), each consisting of three items. The ten items of the WAMI are rated on a 5-point Likert scale, ranging from 1 (absolutely untrue) to 5 (absolutely true). For our sample, we used the Romanian version of the WAMI translated from English through a translation–backtranslation process. The greater good motivations subscale was subsequently removed from the analysis since it did not meet the minimum criteria for internal consistency (Cronbach’s Alpha = 0.345 < 0.7).

Exhaustion was measured using the Exhaustion scale from The Maslach Burnout Inventory (MBI) [50], which was validated on the Romanian population [51]. The Exhaustion scale consists of five items (e.g., “I feel used up at the end of the workday.”) rated on a 7-point Likert scale ranging from 0 (never) to 6 (every day).

Job demands were measured using scales from the Job Demands-Resources Questionnaire (JD-RQ) [52] translated into Romanian through a translation–backtranslation process. Four scales were selected: work pressure (e.g., “Do you work under time pressure?”), composed of four items; emotional demands (e.g., “Do you face emotionally charged situations in your work?”), composed of six items; role conflict (e.g., “At my work, different groups of people expect opposite things from me”), composed of four items; and hassles (e.g., “I have many hassles to go through to get projects/assignments done”), composed of five items. Each scale was measured on a 5-point Likert scale ranging from 1 (never) to 5 (very often) or 1 (strongly disagree) to 5 (strongly agree).

We executed a Confirmatory Factor Analysis (CFA) in order to establish whether challenging demands (work pressure and emotional demands) and hindering demands (role conflict and hassles) covered two different constructs. We found that the 2-factor model showed better fit indices than the 1-factor model, which supported the hypothesized challenging–hindering demands factor structure. The SRMR value for the 2-factor model was substantially lower than for the 1-factor model (SRMR2f = 0.008 < SRMR1f = 0.067), while the NFI value was larger for the 2-factor model (NFI2f = 0.997 > NFI1f = 0.876) [53].

### 2.3. Data Analysis

All statistical analyses were performed using JASP (v. 0.13.1) [54] and SmartPLS (v 3.3.3) [55]. First, participants with missing values on one or more of the used scale items were removed from the analysis. In the next step, participants with inconsistent responses were also removed, thus reducing the size of the valid sample to *n* = 1096. Moderation analyses were performed using a partial least squares (PLS)-based SEM modeling, given that our data presented a significant difference from normality. Our proposed model is a Higher Order Construct (HOC) model and consists of three 2nd order formative constructs, challenging demands, hindering demands, and work meaning, and seven first-order reflective constructs, with Exhaustion being the endogenous variable. The conceptual model can be seen in Figure 1. For good model fit, we considered values of SRMR (standardized root mean square residual) to be less than 0.08 and values of NFI (Bentler–Bonett Normed Fit Index) to be greater than 0.9 [53]. To test the significance of the path coefficients and loadings, we used the bootstrapping method. The starting number of bootstrap samples used for analysis was 1500, as referenced by the authors of [56], for small rejection probabilities of the null hypothesis. Further increases in the number of bootstrapping samples above the 5000 mark did not produce significant changes in the results.

## 3. Results

### 3.1. Descriptive Statistics

The descriptive statistics, including the means, standard deviations, and correlations of the study variables can be found in Table 1.

### 3.2. Measurement Model

In order to assess the measurement model, we used the joint two-stage approach [57]. First, we assessed the first-order constructs in the framework for convergent validity, taking into consideration factor loadings, Cronbach’s Alpha (CA), Composite Reliability (CR), and Average Variance Extracted (AVE). Following this, two items reflecting the hassles construct were removed because the loadings were smaller than the 0.6 threshold [58]. This operation increased the construct’s AVE value. Table 2 shows that with these modifications, the measurement model met the quality criteria (loadings > 0.6, CA > 0.7, CR > 0.7, AVE > 0.5).

Discriminant validity was evaluated next, using the heterotrait–monotrait (HTMT) ratio of correlations. All values obtained were acceptable according to the HTMT inference criteria [59] and are shown in Table 3.

Finally, we used PLS and bootstrap path analyses to assess the second-order constructs by examining the corresponding first-order constructs’ weights/loadings and the Variable Inflation Factors (VIF). To do this, we generated a new path model based on the latent variable scores obtained in the first step. Table 4 shows that all formative indicators were adequate and that no collinearity issues were detected (VIF < 2.5).

### 3.3. Structural Model

The final model is based on the latent variable scores generated in the initial analysis. The model showed a good fit to the data in relation to our proposed reference values (SRMR = 0.020 < 0.08, NFI = 0.987 > 0.9). For model assessment, we considered the R-squared (R^2^) and beta (β), along with corresponding *p*-values and the predictive relevance (Q^2^) [58], as suggested by the authors of [60] (see Figure 2).

Assessing the relationships between variables, the path analysis showed that both challenging demands (β = 0.311, ρ < 0.001) and hindering demands (β = 0.216, ρ < 0.001) positively predicted exhaustion (H1). Higher perceived intensity of job demands predicted higher emotional exhaustion levels in firefighters. In addition, work meaning had a significant negative relationship with exhaustion (H2) (β = −0.270, ρ < 0.001), indicating that the more meaning one finds in work, the lower the risk of negative strain. Overall, the model accounted for 30.4% of the variance in exhaustion (R^2^ = 0.304). Following the PLS blindfolding procedure, a predictive relevance value of 0.295 (Q^2^ > 0) was extracted for the endogenous variable exhaustion, indicating that the model had acceptable predictive relevance.

### 3.4. Moderation Analysis

In order to test the moderation effect of work meaning, two additional interaction constructs (work meaning x challenging demands; work meaning x hindering demands) obtained via the PLS product-indicator approach were added to the model. The moderation analysis revealed that work meaning x challenging demands had a small but significant negative effect (β = −0.088, ρ = 0.014) (H3) and introduced 0.9% additional variance in the model. The effect size was small (f^2^ = 0.016). However, the second effect, work meaning x hindering demands, was also significant (β = −0.106, ρ = 0.002) (H3) and larger in comparison with the first effect. The addition of the second interaction term caused a 1.4% increase in R^2^ and produced a small effect size (f^2^ = 0.024). These findings offer support for the potential role of work meaning in decreasing the impact of job demands on emotional exhaustion. The slope analysis (see Figure 3) suggests that for higher levels of work meaning, firefighters in our sample reported experiencing lower levels of exhaustion as a consequence of job demands than for lower levels of work meaning. Table 5 shows the hypotheses summary.

## 4. Discussion

The aim of this study was to assess the buffering role of work meaning in the “health impairment” process of the JD-R theory, in which job demands are related to higher exhaustion levels, in the context of firefighters’ work. We argued that work meaning plays an important role in firefighters’ work and sense of professional identity and that experiencing meaning in work allows for better management of job demands and associated stress. This research adds to the existing literature on the JD-R theory and exhaustion by assessing the moderating effect of work meaning between job demands and exhaustion levels in firefighters.

### 4.1. Contributions

First, the results showed a positive association between job demands and exhaustion. This finding is in accordance with the propositions of the JD-R model’s “health impairment” process and the results of other studies [2,61]. Our data revealed a slightly larger effect for challenging job demands (work pressure and emotional demands) against hindering job demands (role conflict and hassles) in predicting emotional exhaustion levels among firefighters. This could be explained by the health impairment potential that requirements specific to firefighters’ missions have. According to [2], heavy physical activity, time pressures, and emotionally demanding situations in the workplace were associated with emotional exhaustion in a sample of 101 French firefighters. The study concluded that the work of firefighters appears to be a strong source of stress and mental strain. On the other hand, another study [62] assessing differences in job characteristics between Polish police officers and firefighters found that tasks performed at the fire station are less absorbing and are not felt as strongly as those required in field operations. In such circumstances, hindering job demands, such as hassles or role conflicts in the workplace, may become secondary yet remain significant sources of potential strain even in the case of military structures, such as the firefighting service.

Second, it was found that firefighters with higher levels of work meaning reported less emotional exhaustion. Consistent with findings in other studies mentioned in this paper, this result suggests that finding a positive meaning in work and drawing a sense of personal growth, purpose, and a deeper understanding of life from one’s work experiences is associated with a lower risk of developing burnout symptoms. The pursuit of meaning is fundamental in achieving an adaptive mode of orientation in the environment, a forward movement toward desired goals, and the maintenance of cohesive and stable self-states. The workplace is often a major scene for such dynamics to be acted out. Work meaning seems to be a valuable psychological resource that can enable the use of personal strengths and a degree of resilience in the face of adversity. Deficits in this area might determine detached or impersonal attitudes toward various aspects of work, proneness to emotional and physical exhaustion, disengagement, and a sense of discontinuity in one’s self-experience. Finding meaning in work is especially important for firefighters since their profession implies high demands in terms of resilience to stressful situations [63,64,65,66].

In addition, our data indicate the existence of a moderating effect introduced by work meaning. In accordance with our hypothesis, it was found that work meaning buffered the impact of firefighters’ job demands on emotional exhaustion levels. The higher the sense of meaning associated with work, the lower the risk of developing exhaustion due to challenging and hindering job demands. This result suggests that firefighters who derive personal meaning and existential significance from their work are more likely to effectively engage in work and better manage job demands and negative strain. To our knowledge, this is the first study directly assessing the interaction between work meaning and job-related stress-inducing factors. It is worth noting that in our sample, the moderating effect was stronger in relation to hindering job demands, such as administrative hassles and conflicting expectations in the workplace. It is possible that a strong sense of meaning enables individuals to be more engaged [37,43,67,68] and maintain their focus on the positive and significant aspects of their activity rather than getting stuck on various bureaucratical or procedural inconsistencies and contradictory demands. Nevertheless, our data suggest that even in the case of a military structure, which entails rigid rules, regulations, and orders, administrative hassles and conflicting expectations present significant threats to personnel performance and well-being. As such, this result shows potential promise for future research regarding the prevention of burnout symptoms in firefighters and the JD-R gain–loss dynamics in general. The impact of emotionally charged situations and work-related pressure on exhaustion levels in firefighters was also buffered by work meaning but to a lesser degree. However, the effect was significant, revealing that work meaning is a contributing factor to resilience and mental stability within a highly demanding work environment. Still, the small effect sizes obtained in this study raise questions about the practical significance of these findings, and future studies are needed to assess confidence in this buffering effect.

### 4.2. Limitations

A number of limitations in the present study should be noted. First, the cross-sectional approach we used in this study does not allow for conclusions regarding causality or the circular effects that the variables may induce. A longitudinal research design should offer the opportunity for better insight on dynamics concerning the role of work meaning among firefighters. Another issue concerns the potential bias in the data given the use of self-reported measures. Second, the results of this research were obtained within the specific cultural, economic, and institutional landscape of Romania. Hence, comparison with other firefighter populations from different cultures with different values and working conditions is needed in order to explore the general or particular nature of these results. Third, given the lack of consensus in the research field on how work meaning should be defined and measured, the use of only one instrument may fail to capture relevant dimensions of this concept. In addition, future studies could be aimed at better understanding how work meaning relates to higher-order concepts, such as meaning in life, to increase the resolution on specific contributions in preventing and alleviating the physical and mental consequences of prolonged exposure to stress in the workplace.

### 4.3. Practical Implications

The firefighting profession is inherently stressful [1]. The harmful potential of the specific stressors has been extensively documented [69]. Various other factors have proven to have a buffering effect on burnout dimensions (e.g., social support, lifestyle practices, leadership behaviors, and work engagement) but, to our knowledge, not work meaning. The results of this study, although showing significant but small effects, bring new practical directions toward preventing burnout symptoms in firefighters.

First, since nothing significant can be done regarding the intrinsic nature of the firefighting profession (challenging job demands), we cannot say the same concerning hindering job demands. As such, hassles, role conflicts, and excessive bureaucracy depend on organizational efforts to improve. By taking action at the management level, the leading forces can be made aware of this challenge and conceive an action plan for improvement. Second, new psychological intervention directions are available now in work meaning actions. For instance, psychological training and preparedness before missions could address work meaning issues, emphasizing the positive meaning of their work using concrete aspects/examples. In addition, debriefing sessions carried out after complex interventions could add more significance to the completed mission. In this regard, mass-media analysis and public opinions can be used as practical materials in constructing and developing meaning. Controlled group sessions could be favorable contexts for creating meaning through the flow of ideas and experiences related to various aspects of firefighters’ work and work-related experiences. For instance, our previous research [37] found that firefighters actively engage in job crafting and consequently shape the meaning that work has in their lives. In conclusion, either through organizational or psychological means, actions taken toward deriving personal significance from work and a broader sense of meaning should be addressed synergistically to aim to prevent burnout symptoms. Future research might assess the effects of such practical interventions on work meaning and burnout.

## 5. Conclusions

In summary, we replicated some of the previous findings concerning relationships between job demands, work meaning, and emotional exhaustion, and we also introduced an additional interaction factor in explaining the variance of exhaustion levels in firefighters. Our study found that individuals who perceived job demands as more intense were more likely to exhibit higher levels of emotional exhaustion. Conversely, deriving personal meaning from work was associated with lower levels of exhaustion in firefighters. According to our results, work meaning (positive meaning and meaning making) presented a small but significant buffering effect on the impact of job demands (challenging and hindering) on emotional exhaustion. Therefore, we recommend a cautious stance in interpreting this result, and we encourage further replication and research on this topic.

## Figures and Tables

**Figure 1 ijerph-18-09819-f001:**
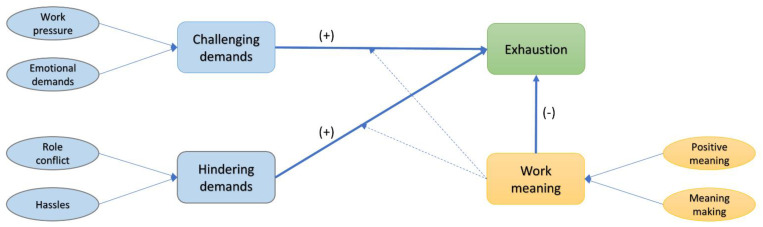
Hypothesized relationships of the research model.

**Figure 2 ijerph-18-09819-f002:**
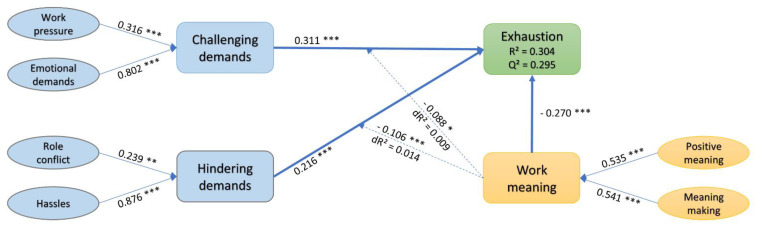
Results of the research model. Path estimates were significant (* ρ < 0.05, ** ρ < 0.01, *** ρ < 0.001) and reported as standardized.

**Figure 3 ijerph-18-09819-f003:**
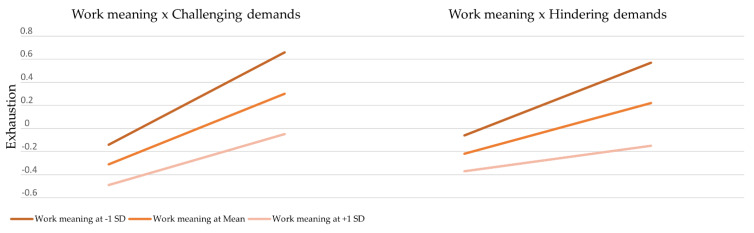
Interaction between work meaning and job demands predicting exhaustion.

**Table 1 ijerph-18-09819-t001:** Descriptive statistics and inter-correlations of the study variables.

	M	SD	1	2	3	4	5	6
1. work pressure	11.4	3.08	-					
2. emotional demands	14.1	4.62	0.505 ***	-				
3. role conflict	9.44	2.59	0.157 ***	0.234 ***	-			
4. hassles	14.4	3.63	0.304 ***	0.367 ***	0.381 ***	-		
5. positive meaning	16.4	2.17	0.003	−0.062 *	−0.179 ***	−0.158 ***	-	
6. meaning making	12	1.80	0.018	−0.036	−0.091 **	−0.109 ***	0.711 ***	-
7. Exhaustion	4.16	4.35	0.287 ***	0.392 ***	0.241 ***	0.344 ***	−0.306 ***	−0.301 ***

Note. * ρ < 0.05, ** ρ < 0.01, *** ρ < 0.001.

**Table 2 ijerph-18-09819-t002:** Assessment of reflective constructs.

	CA	CR	AVE	Item Loadings					
				1	2	3	4	5	6
Challenging demands									
work pressure	0.710	0.821	0.536	0.685	0.774	0.665	0.795	-	-
emotional demands	0.848	0.888	0.572	0.786	0.726	0.825	0.784	0.787	0.611
Hindering demands									
role conflict	0.797	0.805	0.623	0.724	0.752	0.851	0.823	-	-
Hassles	0.822	0.822	0.737	0.837	0.861	0.878	-	-	-
Work meaning									
positive meaning	0.788	0.791	0.612	0.774	0.749	0.801	0.803	-	-
meaning making	0.758	0.757	0.674	0.779	0.846	0.837	-	-	-
Exhaustion	0.897	0.902	0.709	0.823	0.789	0.87	0.87	0.855	-

Note. CA—Cronbach’s Alpha, CR—Composite Reliability, AVE—Average Variance Extracted.

**Table 3 ijerph-18-09819-t003:** Discriminant validity.

	1	2	3	4	5	6
1. work pressure	-					
2. emotional demands	0.643	-				
3. role conflict	0.218	0.286	-			
4. hassles	0.411	0.455	0.51	-		
5. positive meaning	0.108	0.084	0.23	0.278	-	
6. meaning making	0.087	0.091	0.123	0.193	0.93 *	-
7. Exhaustion	0.362	0.448	0.285	0.452	0.364	0.369

Note. * CI [0.883;0.978]—includes HTMT.90 criteria.

**Table 4 ijerph-18-09819-t004:** Quality assessment for formative constructs.

	Weights	Loadings	VIF
work pressure -> Challenging demands	0.316 ***	0.723 ***	1.348
emotional demands -> Challenging demands	0.802 ***	0.962 ***	1.348
role conflict -> Hindering demands	0.239 **	0.605 ***	1.210
hassles -> Hindering demands	0.876 ***	0.973 ***	1.210
positive meaning -> Work meaning	0.535 ***	0.929 ***	2.136
meaning making -> Work meaning	0.541 ***	0.925 ***	2.136

Note. ** ρ < 0.01, *** ρ < 0.001, VIF—Variance Inflation Factor.

**Table 5 ijerph-18-09819-t005:** Hypotheses summary.

Hypothesis	Relationship	β	f^2^	Decision
H1	Challenging demands -> Exhaustion	0.311 ***	0.115	Supported
	Hindering demands -> Exhaustion	0.216 ***	0.053	
H2	Work meaning -> Exhaustion	−0.270 ***	0.100	Supported
H3	Work meaning x Challenging demands -> Exhaustion	−0.088 *	0.016	Supported
H4	Work meaning x Hindering demands -> Exhaustion	−0.106 ***	0.024	Supported

Note. * ρ < 0.05, *** ρ < 0.001, β—standardized path coefficient, f^2^—Cohen’s f squared.

## Data Availability

The datasets generated for this study will not be made publicly available because of institutional constraints. Requests to access the datasets should be directed at cristinaioanadan@yahoo.com.

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
