# Peer review of "Job Demands and Exhaustion in Firefighters: The Moderating Role of Work Meaning. A Cross-Sectional Study"

_ijerph, 2021, doi:10.3390/ijerph18189819_

Round 1
Reviewer 1 Report
Dear authors and editors,
first of all I want to congratulate authors for their manuscript and for very interesting study you have conducted in Romania.
This study is very important for the field of workplace stress and influence to health. It is very significant finding of work meaning protective role to emotional exhaustion. The study strength is multicentre design and representative sample. This study will certainly give add value to science and significant contribution for prevention measures. For future research should be reconsider longitudinal design and qualitative approach.
For this manuscript my suggestions were to add study design in title, not use abbreviations in abstract and to better connect international results with results of this study, connect with tables and figures.
Title is usually a clear descriptive account of your work. It should indicate method used. I suggest to add study design.
Do not use abbreviations in abstract and in key words ( JD-R)
Introduction - well done
Methods and Results - clearly presented
Please connect your findings (tables) with international research.
Reviewer 2 Report
There are just a few minor English/grammatical issues that could be fixed with careful proofreading. For example, in the Abstract (line 18), this should read, “has not been studied” rather than “was not studied”. On p. 4 (line 167), instead of “retroversion”, do you mean translation-backtranslation process? Retroversion isn’t clear.
Clarify the form of the interaction in the Abstract (lines 24-25). What is meant by an interaction “negatively predicting exhaustion” is not clear. For example, if the interaction term was negative, then did higher personal meaning attenuate the positive relationship between demands and exhaustion? (Note: the following sentence in the Abstract is actually clearer, so perhaps just use that.)
Given that previous research (p. 2, lines 46-48) has already demonstrated that meaning buffers the impact of work-related stress, why would you expect these relationships to differ in this population? Obviously, you don’t ? But it does highlight that this study is primarily a replication of prior research with a slight extension by virtue of the sample used.
Hypothesis 3 poses a buffering effect of meaning on both challenging and hindering demands. Yet, just above (lines 115-117), the authors note that challenging demands can lead to positive emotions, engagement, growth and mastery. So, for challenging demands, wouldn’t you expect that personal meaning would only magnify (rather than attenuate) those beneficial outcomes? (I am fine with the way the buffering interaction is predicted for hindering demands, but that doesn’t seem to comport with what you stated about challenging demands).
On that note, I think your hypotheses should better clarify which challenge and hindrance demands you are looking at in this paper specifically, since there are many possibilities.
Was the sample of firefighters representative of the population? For example, only .6% were women; is this also true of the population of Romanian firefighters? (I expect so, but would be good for the reader to evaluate this and other demographic characteristics to the extent possible.)
The description of “reflective-formative type constructs” (p. 4, lines 197-198) is unclear. Constructs are either reflective OR formative; they are not both. So, please clarify the direction of the arrows for the relationships between items and constructs (i.e., items reflect an underlying latent construct OR items form the construct). Also, if some were reflective and others formative, which were which?
Overall, this is very nicely written and makes a modest contribution to the extant literature. As the authors themselves note, the effect sizes associated with the interaction terms were quite low which calls into question the practical importance of the results. Nevertheless, the study was soundly conducted and the authors acknowledged the inherent limitations.
Reviewer 3 Report
In the present paper, the authors examine the role of work meaning in the the relationship between job demands and related emotional exhaustion. To my view, the general research goal is a highly valuable one. There is much to like about the paper. It is well-written, it reports a set of four coherent hypotheses conceptually well justified, conclusions are appropriate and limitations well reported, and authors investigate a wide sample of firefighters. However, I think the authors can improve the paper in minor details. For example, being cautious with the aim of the study. In the abstract, for example, "to examine the buffering role of work meaning" is difficult to do with a cross-sectional study. Authors could reformulate the aims to adecuate them to a cross-sectional study. I recommend to check and review tables and figures, to maintain coherence between them and with style. My sincere congratulations for the article!
